# A general learning scheme for classical and quantum Ising machines

**Ludwig Schmid[1]⋆, Enrico Zardini[2] and Davide Pastorello[3,4]**

**1** Chair for Design Automation — Technical University of Munich, 80333 Munich, Germany
**2** Department of Information Engineering and Computer Science —
University of Trento, 38123 Trento, Italy
**3** Department of Mathematics — University of Bologna, 40126 Bologna, Italy
**4** TIFPA-INFN, 38123 Povo-Trento, Italy

⋆ ludwig.s.schmid@tum.de

## Abstract

An Ising machine is any hardware specifically designed for finding the ground state of the Ising model. Relevant examples are coherent Ising machines and quantum annealers. In this paper, we propose a new machine learning model that is based on the Ising structure and can be efficiently trained using gradient descent. We provide a mathematical characterization of the training process, which is based upon optimizing a loss function whose partial derivatives are not explicitly calculated but estimated by the Ising machine itself. Moreover, we present some experimental results on the training and execution of the proposed learning model. These results point out new possibilities offered by Ising machines for different learning tasks. In particular, in the quantum realm, the quantum resources are used for both the execution and the training of the model, providing a promising perspective in quantum machine learning.

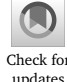

## 1   Introduction

Machine learning models are algorithms that provide predictions about observed phenomena by extracting information from a set of collected data (the training set). In particular, *parametric models* capture all relevant information within a finite set of parameters, with the set being independent of the number of training instances [1]. A celebrated example is represented by *artificial neural networks* [2–4]. In the context of quantum computers, a common approach to machine learning is to employ variational quantum circuits, which can be trained by backpropagation as done with classical feedforward neural networks [5–8]. In addition to gate-based quantum computing, *quantum annealing* has also been considered to develop machine learning algorithms [9–11]. In any case, a crucial point in quantum machine learning is the implementation of quantum procedures for model training as alternatives to classical methods. An example in this sense is the quantum support vector machine, trained by running the HHL quantum algorithm [12], which, however, presents the shortcoming of an impractical implementation on the currently available quantum devices. Therefore, a general challenge in quantum machine learning is to define learning schemes that can be efficiently implemented on quantum machines of the Noisy Intermediate-Scale Quantum (NISQ) era [13]. This is the motivation behind the present proposal of a learning model for quantum annealers in which the quantum resources are used both in the model execution and in the training process. The obtained theoretical and experimental results apply also to classical implementations of the model. Indeed, the key aspect of the training and execution of the proposed learning mechanism is the computation of the ground state of the *Ising model*, which can, in principle, be solved using classical or quantum procedures.

An *Ising machine* can be considered a specific-purpose computer designed to return the absolute or approximate ground state of the Ising model. The latter is described by the energy function of a spin glass system under the action of an external field, namely,

$$E(\mathbf{z}) = \sum_{i=1}^{N} \theta_i z_i + \sum_{(i,j)} \Gamma_{ij} z_i z_j, \quad \text{with} \quad \mathbf{z} \in \{-1,1\}^N, \quad \theta_i \in \mathbb{R}, \quad \text{and} \quad \Gamma_{ij} \in \mathbb{R}, \qquad (1)$$

where the sum $\sum_{(i,j)}$ is taken over the pairs of connected spins, counting each pair only once. The ground state is the spin configuration $\mathbf{z}^* \in \{-1,1\}^N$ that minimizes the function (1). Therefore, in practice, an Ising machine solves a combinatorial optimization problem that can be represented as a quadratic unconstrained binary optimization (QUBO) problem, which is an NP-hard problem, by means of the change of variables $x_i = \frac{z_i+1}{2} \in \{0,1\}$. In particular, an Ising machine can be an analog computer that evolves toward the Ising ground state due to a physical process like thermal or quantum annealing. Alternatively, it can also be implemented on a digital computer in terms of simulated annealing.

Ising machines are conceptually related to *Boltzmann machines* in the sense that they are both defined in terms of the Ising model, with couplings among spins and the action of an external field. In the case of a Boltzmann machine, the coefficients $\theta$ and $\Gamma$ of the energy function (1) are tuned so that, by sampling the spin configuration over the state of the system at thermal equilibrium (at a finite temperature $T$), a probability distribution resembling an input distribution defined on the training set is generated [14].

In detail, the output distribution of a Boltzmann machine is given by

$$p_T(\mathbf{z}) = Z^{-1} \exp\left[-\frac{\mathsf{E}(\mathbf{z})}{k_B T}\right], \tag{2}$$

where $Z := \sum_{\mathbf{z}} \exp\left[-\frac{\mathsf{E}(\mathbf{z})}{k_B T}\right]$ is the partition function and $k_B$ is the Boltzmann constant. Usually, only a subset of spins is sampled, the so-called *visible nodes*, and the output distribution is given by the marginal distribution of (2). Instead, in the ideal case, the output of an Ising machine is deterministic and corresponds to the absolute minimum of (1). However, in a realistic scenario in which the Ising machine operates by thermal annealing, the output is probabilistic and distributed according to (2) with a value of $T$ as low as possible.

The difference between Boltzmann and Ising machines lies in the fact that Boltzmann machines are parametric generative models. In contrast, Ising machines are considered as solvers of combinatorial optimization problems [15–17]. However, in this paper, we propose a supervised learning model for Ising machines whose training is inspired by the training of Boltzmann machines. A peculiar aspect of a Boltzmann machine is that it can be trained by gradient descent of a loss function $\mathcal{L}$ depending on the weights $\theta$ and $\Gamma$, like the average negative log-likelihood between the input distribution and the generated distribution, iteratively changing the parameters by a step in the opposite direction of the gradient. However, the partial derivatives of $\mathcal{L}$ are not explicitly calculated but are estimated by sampling the network units. For instance, let us consider the update rule $\Gamma_{ij} \to \Gamma_{ij} + \delta\Gamma_{ij}$, which updates the coupling terms toward the minimum of the average negative log-likelihood. The update step ($\delta\Gamma_{ij}$) is given by [14]:

$$\delta\Gamma_{ij} = -\eta\left(\langle z_i z_j\rangle - \sum_{\mathbf{v}} p_{data}(\mathbf{v})\langle z_i z_j\rangle_{\mathbf{v}}\right), \quad i,j = 1, ..., N, \tag{3}$$

where $\eta > 0$ is the learning rate (user-specified), the sum is taken over the visible nodes $\mathbf{v}$, $p_{data}$ is the input distribution, $\langle\ \rangle$ is the Boltzmann average, and $\langle\ \rangle_{\mathbf{v}}$ is the Boltzmann average with clamped visible nodes. In other words, both the training and the execution of a Boltzmann machine are performed by sampling the units of the network at thermal equilibrium. A quantum version of the Boltzmann machine has also been proposed [18], and the simulations have shown that the presence of a *transverse field Hamiltonian* improves the training process with respect to the classical model, generating distributions that are closer to the input one in terms of the Kullback-Liebler divergence.

This paper adopts a similar viewpoint for training an Ising machine. After defining a parametric predictive model based on the ground state of the Ising model, we prove that it can be trained by gradient descent of a mean squared error loss function, executing the model itself to obtain the gradient estimates. In particular, the structure of the model does not require that the Ising machine returns the true ground state with infinite precision, and a suboptimal output works for training and executing the predictive model. In addition, our results apply to both classical and quantum machines. However, in the second case, the impact may be more significant since the quantum annealing resources are also exploited for the training process. In this sense, the purpose is similar to that of the *parameter-shift rule*, which is used in gate-based quantum computing to train a parametric quantum circuit without explicitly calculating the partial derivatives [19].

The paper is structured as follows: in Section 2, we introduce generalities and elementary notions about the Ising model and Ising machines, with a particular focus on quantum annealing; Section 3 deals with the proposed parametric learning model, to be executed by an Ising machine, and the main theoretical result of the paper, i.e., the proof that the model can be trained by running the Ising machine itself; in Section 4, an empirical evaluation of the proposed machine learning method is provided; in Section 5, we discuss the perspectives of the proposal, and we draw our conclusions on the proposed parametric model.

## 2 Ising machines

This section introduces the formal definition of the Ising model and the concept of using specific Ising machines to solve the corresponding groundstate problem. Afterward, we briefly describe the two Ising machines employed in this work, namely simulated and quantum annealing.

The *Ising model* is a mathematical description extensively utilized in the study of ferromagnetism. Renowned for its versatility and simplicity, it stands as a fundamental paradigm in the domain of statistical mechanics [20]. In its general formulation, the Ising model is defined on a graph $(V, E)$, wherein each vertex represents a discrete variable $z_i \in \{-1, 1\}$. These variables correspond to *spins*, with associated *biases* $\theta_i \in \mathbb{R}$ denoting the inclination of each spin toward one of the two available values. Furthermore, the weighted edges $\Gamma_{ij} \in \mathbb{R}$ connecting two spins $i$ and $j$ define the coupling dynamics between the spins, indicating their preference to align or oppose each other in value. This graph structure is illustrated in Figure 1. The total energy of a spin configuration $\mathbf{z} \in \{-1, 1\}^{|V|}$ is expressed as

$$\mathsf{E}(\theta, \Gamma, \mathbf{z}) = \sum_{i=1}^{|V|} \theta_i z_i + \sum_{(i,j) \in E} \Gamma_{ij} z_i z_j \ = \theta \mathbf{z} + \mathbf{z}^T \Gamma \mathbf{z}, \tag{4}$$

where the *biases* $\theta_1, ..., \theta_{|V|} \in \mathbb{R}$ and the *couplings* $\Gamma_{ij} \in \mathbb{R} \ \forall (i, j) \in E$ are conveniently consolidated into the vector $\theta$ and the matrix $\Gamma$ (with $\Gamma_{ij} = 0$ when $(i, j) \notin E$), respectively. Realistically, the values of the parameters are bounded. Hence, it is possible to assume that biases and couplings take values into compact intervals of $\mathbb{R}$. Within the realm of statistical physics, these quantities are typically referred to as the external magnetic field strength and spin interactions due to their fundamental roles in the physical manifestation of the Ising model.

An *Ising machine* can be defined as a non-von Neumann computer for solving combinatorial optimization problems [21]. More precisely, its input is represented by the energy function of the Ising model (4), with biases and coupling terms properly initialized. The machine effectively operates by minimizing the energy function and providing the optimal spin configuration $\mathbf{z}^*$ as the output. Actually, the quest to determine the ground state of an Ising model is of significant importance, as any problem within the NP complexity class can be formulated as an Ising problem with only a polynomial increase in complexity [22]. An elementary and abstract definition of an Ising machine, motivated by the general approach adopted in this paper, is the following:

**Definition 1.** *Given the energy function defined in (4), an* **(abstract) Ising machine** *is any map* $(\theta, \Gamma) \mapsto \mathbf{z}^* := argmin_{\mathbf{z}} \mathsf{E}(\theta, \Gamma, \mathbf{z})$.

Additionally, we can also consider the minimum value of the energy $\mathsf{E}_0(\theta, \Gamma) := \mathsf{E}(\theta, \Gamma, \mathbf{z}^*)$ as the output of an Ising machine. This ground state energy of the Ising model is obtained by substituting the spin configuration $\mathbf{z}^* = \mathrm{argmin}_{\mathbf{z}} \mathsf{E}(\theta, \Gamma, \mathbf{z})$ into (4). In this context, the Ising machine consistently yields a numerical result with a negative sign. An illustration of an Ising machine that finds the ground state of a small Ising model is shown in Figure 1.

Relevant examples of Ising machines as specific-purpose hardware devices are quantum annealers [23] or coherent Ising machines with optical processors [24–27]. However, an Ising machine can also be simulated on a classical digital computer. In this respect, simulated annealing is a standard approach and addresses the Ising model as a combinatorial optimization problem. In more detail, simulated annealing is a probabilistic metaheuristic inspired by the analogical notion of controlling the cooling process observed in physical materials [28]. The algorithm employs stochastic acceptance criteria, resembling a Boltzmann probability, to navigate the solution space and escape local optima. Over time, usually indicated by a temperature parameter $T$ that mimics the cooling process, less favorable moves are increasingly rejected.

Figure 1: **Ising model and Ising machine:** On the left, an illustration of the graph structure of an Ising model characterized by a fully connected graph, with $|V| = 5$ spins $\mathbf{z}$, corresponding biases $\theta$, and couplings $\Gamma$. An Ising machine maps the Ising model to the right-hand side of the figure, returning a $\{-1, +1\}$ assignment (illustrated as white/black nodes) to each binary variable $z_i$. The output is the spin configuration $\mathbf{z}^*$ and the corresponding minimal energy $\mathsf{E}_0(\theta, \Gamma)$.

In practice, simulated annealing employs random search and local exploration to converge toward near-optimal or optimal solutions. However, although the algorithm is easy to implement and robust from a theoretical point of view, it may present a slow convergence rate [29]. A promising alternative path is the development of analog platforms like coherent Ising machines. They represent optical parametric oscillator (OPO) networks in which the collective mode of oscillation beyond a certain threshold corresponds to an optimal solution for a given large-scale Ising model [24–27]. The learning scheme proposed here is agnostic and can be implemented on this kind of Ising machines. Nevertheless, in the experimental part we have considered only simulated and quantum annealing.

Quantum annealing is a type of heuristic search used to solve optimization problems [23, 30–32]. The procedure is implemented by the time evolution of a quantum system toward the ground state of a *problem Hamiltonian*. More precisely, let us consider the time-dependent Hamiltonian

$$H(t) = \gamma(t)H_D + H_P, \quad t \geq 0, \tag{5}$$

where $H_P$ is the problem Hamiltonian, $H_D$ is the *transverse field Hamiltonian*, and $\gamma : \mathbb{R}^+ \to \mathbb{R}$ is a decreasing function. Roughly speaking, $H_D$ gives the kinetic term inducing the exploration of the solution landscape by means of quantum fluctuations, and $\gamma$ attenuates the kinetic term driving the system toward the ground state of $H_P$. Quantum annealing can be physically realized by considering a network of qubits arranged on the vertices of a graph $(V, E)$, with $|V| = n$ and whose edges $E$ represent the couplings among the qubits. In detail, the problem Hamiltonian is defined as the following self-adjoint operator on the $n$-qubit Hilbert space $\mathsf{H} = (\mathbb{C}^2)^{\otimes n}$:

$$H_P = \sum_{i \in V} \theta_i \sigma_z^{(i)} + \sum_{(i,j) \in E} \Gamma_{ij} \sigma_z^{(i)} \sigma_z^{(j)}, \tag{6}$$

with real coefficients $\theta_i, \Gamma_{ij}$, which are identified again as biases and couplings due to their similar role in the Ising model. In the computational basis, the $2^n \times 2^n$ matrix $\sigma_z^{(i)}$ acts locally as the Pauli matrix

$$\sigma_z = \begin{pmatrix} 1 & 0 \\ 0 & -1 \end{pmatrix}, \tag{7}$$

on the $i$-th tensor factor and as the $2 \times 2$ identity matrix on the other tensor factors. In fact, the eigenvectors of $H_P$ form the computational basis of $\mathsf{H}$, and the corresponding eigenvalues are the values of the classical energy function (4). On the other hand, for the transverse field Hamiltonian a typical form is

$$H_D = \sum_{i \in V} \theta_i \sigma_x^{(i)}, \tag{8}$$

where the local operator $\sigma_x^{(i)}$ is defined in a similar way to $\sigma_z^{(i)}$ in terms of the Pauli matrix

$$\sigma_x = \begin{pmatrix} 0 & 1 \\ 1 & 0 \end{pmatrix}.$$

$H_D$ does not commute with $H_P$ and provides the unbiased superposition of all the conceivable solutions as the system initial state. Eventually, it is worth highlighting that quantum annealing is related to *adiabatic quantum computing* (AQC) as the solution of a given problem can be encoded into the ground state of a problem Hamiltonian. However, the two notions do not coincide. Indeed, in quantum annealing, the quantum system is not assumed to be isolated; therefore, it can be characterized by a non-unitary evolution. Another difference is that, in quantum annealing, the entire computation is not required to take place in the instantaneous ground state of the time-varying Hamiltonian like in AQC [32].

## 3 The proposed model

This section formally introduces the proposed parametric model, followed by an in-depth discussion on the training using gradient descent and the estimation of the relevant partial derivatives of a quadratic loss function. The final part presents practical considerations required to operate and train the model in real-world scenarios and a discussion of its computational cost.

### 3.1 Definition

In the context of supervised learning, the goal of an algorithm is to approximate a function $f : X \rightarrow Y$ given a training set $\{(x_1, f(x_1)), ..., (x_N, f(x_N))\}$, which is a collection of elements in the set $X$ with the corresponding values of $f$. An approximation of $f$ can be obtained through a parametric function after an optimal choice of its parameters, generalizing the information encoded into the training set. In fact, the notion of a parametric model is closely related to the existence of a parametric function that can be used to approximate the target function.

**Definition 2.** *Let $X$ and $Y$ be non-empty sets respectively called* **input domain** *and* **output domain***. A (deterministic)* **parametric model** *is a function*

$$x \mapsto y = F(x|\Gamma), \quad x \in X, \ y \in Y,$$

*with $\Gamma$ being a set of real parameters.*

In practice, given a training set of input-output pairs, the task consists in finding the parameters $\Gamma$ such that the model assigns the correct or approximately correct output, with high probability, to any previously unseen input. The parameters are typically determined by optimizing a *loss function* such as

$$\mathcal{L}(\Gamma) = \frac{1}{N} \sum_{i=1}^{N} \mathrm{d}(y_i, F(x_i|\Gamma)),$$

where d is a metric defined over $Y$, and the procedure is commonly referred to as *training*.

A preliminary depiction of the general problem considered in this paper is the following: given a real-valued function $f : X \rightarrow \mathbb{R}$, with $X \subset \mathbb{R}^n$ and $n \in \mathbb{N}$, the objective consists in training a predictive model $F$ that approximates the original function $f$ within the supervised learning framework. This function approximation task encompasses a wide range of conventional machine learning endeavors such as regression and classification. In particular, the proposed parametric model is defined over the concept of Ising machines as introduced in Section 2. The input information is encoded into the biases $\theta$ of an Ising model, while the

adjustable parameters are represented by the couplings $\Gamma$ of (4). The Ising machine is then used to find the ground state of the Ising model, and the corresponding ground state energy is used as the model output. Note that the ground state energy invariably assumes a negative value, and the magnitude of the input biases significantly influences its absolute magnitude. To account for this, we introduce an ancillary scaling factor denoted as $\lambda$ and an energy offset indicated as $\epsilon$. This yields the subsequent formulation of the model.

Given an Ising machine, an input vector $\theta = (\theta_1, \ldots, \theta_n) \in X \subset \mathbb{R}^n$, and the parameters $\{\Gamma_{ij}\}$ with $i, j = 1 \ldots n$ (the nonzero $\Gamma_{ij}$ are specified by the topology graph of the machine), one can define a parametric model $F$ based on the ground state energy of an Ising model as

$$
\begin{aligned}
F(\theta|\Gamma, \lambda, \epsilon) &:= \lambda \min_{\mathbf{z} \in \{-1,1\}^n} \mathsf{E}(\theta, \Gamma, \mathbf{z}) + \epsilon \\
&= \lambda \, \mathsf{E}_0(\theta, \Gamma) + \epsilon \,,
\end{aligned}
\tag{9}
$$

where $\lambda \in \mathbb{R}$ and $\epsilon \in \mathbb{R}$ are additional tunable parameters that do not influence the Ising model energy. The model definition reveals a general neural approach in the sense that data are represented by the biases of the spins, which can be associated with neurons, and the parameters are the weights attached to the connections between spins (neurons). It is worth noting that, for the model execution, there is no requirement that the Ising machine returns the true ground state. More precisely, the fact that an approximated ground state does not match the exact solution of the combinatorial problem underlying the minimization is not a severe drawback for the learning process. Indeed, assuming that the deviation of the energy output from $\mathsf{E}_0$ is systematic (e.g., due to the finite precision of the Ising machine), this deviation becomes a characteristic of the model itself, and the training procedure accordingly provides optimized parameters. Despite its simplicity, the model presents interesting training properties that we mathematically characterize in the next section.

## 3.2 Training process

Training the proposed parametric model for the approximation of a real-valued function entails minimizing the empirical risk across a provided dataset, denoted as $\mathcal{D}$, encompassing input-output pairs derived from the original function. To this aim, we employ the conventional approach of optimizing the model parameters to minimize the mean squared error (MSE) between the predicted output and the actual data values.

Given the training set $\mathcal{D} = \{(\theta^{(a)}, y^{(a)})\}_{a=1,\ldots,N}$, with $y^{(a)} = f(\theta^{(a)})$, where $f : X \to \mathbb{R}$, with $X \subset \mathbb{R}^n$, is an unknown function to approximate, the model (9) can be trained by minimizing the MSE loss function

$$
\mathcal{L}(\Gamma, \lambda, \epsilon) = \frac{1}{N} \sum_{a=1}^{N} \left[ F(\theta^{(a)}|\Gamma, \lambda, \epsilon) - y^{(a)} \right]^2 \,.
\tag{10}
$$

Our objective is to address this minimization task employing a gradient descent approach, iteratively updating the parameters $\Gamma$, $\lambda$, and $\epsilon$ by taking steps in the direction opposite to the gradient of the loss function $\mathcal{L}$:

$$
\delta\Gamma = -\eta \nabla_\Gamma \mathcal{L} \,, \qquad \delta\lambda = -\eta \frac{\partial \mathcal{L}}{\partial \lambda} \,, \qquad \delta\epsilon = -\eta \frac{\partial \mathcal{L}}{\partial \epsilon} \,,
\tag{11}
$$

where $\eta > 0$ is the learning rate, which controls the optimization step size. Let us remark that each parameter is assumed to take values into a compact interval in $\mathbb{R}$; consequently, the parameter space is a hyperrectangle. On one hand, the partial derivatives of $\mathcal{L}$ with respect to $\lambda$ and $\epsilon$ are well-defined and trivial to calculate. On the other hand, the following theorem, which provides the update rules for the optimization of $\mathcal{L}$ by gradient descent, implies that the gradient $\nabla_\Gamma \mathcal{L}$ is defined almost everywhere in the parameter hyperrectangle.

**Theorem 3.** *Let $F$ be the parametric model defined in (9), $\mathcal{D} = \{(\theta^{(a)}, y^{(a)})\}_{a=1,...,N}$ be a training set for $F$, $\mathcal{L}$ be the MSE loss function defined in (10), and $\eta > 0$ be the learning rate. Then, the partial derivatives of $F$ with respect to the couplings $\Gamma$ are defined almost everywhere in the parameter space, and the update rules for $\Gamma$, $\lambda$, $\epsilon$ for the gradient descent of $\mathcal{L}$ are:*

$$\Gamma_{ij}^{(k+1)} = \Gamma_{ij}^{(k)} - \eta \frac{2\lambda^{(k)}}{N} \sum_{a=1}^{N} \left[ \lambda^{(k)} \mathsf{E}_0(\theta^{(a)}, \Gamma^{(k)}) + \epsilon^{(k)} - y^{(a)} \right] z_i^* z_j^*, \tag{12}$$

$$\lambda^{(k+1)} = \lambda^{(k)} - \eta \frac{2}{N} \sum_{a=1}^{N} \left[ \lambda^{(k)} \mathsf{E}_0(\theta^{(a)}, \Gamma^{(k)}) + \epsilon^{(k)} - y^{(a)} \right] \left[ \sum_{i=1}^{n} \theta_i^{(a)} z_i^* + \sum_{(i,j) \in E} \Gamma_{ij}^{(k)} z_i^* z_j^* \right], \tag{13}$$

$$\epsilon^{(k+1)} = \epsilon^{(k)} - \eta \frac{2}{N} \sum_{a=1}^{N} \left[ \lambda^{(k)} \mathsf{E}_0(\theta^{(a)}, \Gamma^{(k)}) + \epsilon^{(k)} - y^{(a)} \right], \tag{14}$$

*where $\Gamma^{(k)}$, $\lambda^{(k)}$, $\epsilon^{(k)}$ are the values of the parameters within the $k$-th iteration of the gradient descent, and $\mathbf{z}^* = \mathrm{argmin}_{\mathbf{z}} \mathsf{E}(\theta^{(a)}, \Gamma^{(k)}, \mathbf{z})$.*

*Proof.* By direct calculation, the partial derivative of $F$ with respect to $\Gamma_{ij}$ is

$$\frac{\partial F(\theta | \Gamma, \lambda, \epsilon)}{\partial \Gamma_{ij}} = \lambda \frac{\partial}{\partial \Gamma_{ij}} \left( \sum_{i=1}^{n} \theta_i z_i^* + \sum_{(i,j)} \Gamma_{ij} z_i^* z_j^*, \right) = \lambda z_i^* z_j^*, \tag{15}$$

where $z_i^*$ and $z_j^*$ are the $i$-th and $j$-th components of $\mathbf{z}^* = \mathrm{argmin}_{\mathbf{z}} \mathsf{E}(\theta, \Gamma, \mathbf{z})$, respectively. Since the optimal spin configuration $\mathbf{z}^*$ also depends on $\Gamma$ (and $\theta$), we should consider the derivatives $\partial z_l^* / \partial \Gamma_{ij}$ for $l = 1, ..., n$ in the final step outlined in (15). However, it must be noted that the function $z_l^* = z_l^*(\theta, \Gamma)$ is piecewise constant. Hence, its derivative is zero almost everywhere in its domain, and the remaining points, corresponding to spin flips of $z_l^*$, turn out to be points of non-differentiability of $z_l^*(\theta, \Gamma)$. Substituting (15) into (11), we obtain the following update step ($\delta\Gamma_{ij}$) for the MSE loss function (10):

$$\delta\Gamma_{ij} = -\eta \frac{\partial \mathcal{L}}{\partial \Gamma_{ij}} = -\eta \frac{2}{N} \sum_{a=1}^{N} \left[ F(\theta^{(a)} | \Gamma, \lambda, \epsilon) - y^{(a)} \right] \frac{\partial F}{\partial \Gamma_{ij}} \tag{16}$$

$$= -\eta \frac{2\lambda}{N} \sum_{a=1}^{N} \left[ F(\theta^{(a)} | \Gamma, \lambda, \epsilon) - y^{(a)} \right] z_i^* z_j^*$$

$$= -\eta \frac{2\lambda}{N} \sum_{a=1}^{N} \left[ \lambda \mathsf{E}_0(\theta^{(a)}, \Gamma) + \epsilon - y^{(a)} \right] z_i^* z_j^*. \tag{17}$$

Therefore, the parameter update rule for the $(k+1)$-th iteration turns out to be

$$\Gamma_{ij}^{(k+1)} = \Gamma_{ij}^{(k)} - \eta \frac{2\lambda^{(k)}}{N} \sum_{a=1}^{N} \left[ \lambda^{(k)} \mathsf{E}_0(\theta^{(a)}, \Gamma^{(k)}) + \epsilon^{(k)} - y^{(a)} \right] z_i^* z_j^*, \tag{18}$$

wherein we have omitted the explicit dependence of $z_i^*$ and $z_j^*$ on $a$ and $k$ for the sake of brevity of notation. The update rules for $\lambda$ and $\epsilon$ can be derived analogously. Specifically, the partial derivatives of $F$ with respect to $\lambda$ and $\epsilon$ are

$$\frac{\partial F(\theta | \Gamma, \lambda, \epsilon)}{\partial \lambda} = \sum_{i=1}^{n} \theta_i z_i^* + \sum_{(i,j)} \Gamma_{ij} z_i^* z_j^*, \qquad \frac{\partial F(\theta | \Gamma, \lambda, \epsilon)}{\partial \epsilon} = 1. \tag{19}$$

Then, the claims (13) and (14) follow. $\qquad\square$

---

**Algorithm 1:** Model training

---

    **Input:** dataset $\mathcal{D} = \{(\theta^{(a)}, y^{(a)})\}_{a=1,\dots,N}$, learning rate $\eta$, optimization steps $N_{\text{epochs}}$
    **Output:** trained model $F_{\text{model}}(\theta)$

**1**   Initialize the parameters $\Gamma$;
**2**   **for** step $k$ in $N_{\text{epochs}}$ **do**
**3**      **for** $(\theta^{(a)}, y^{(a)})$ in $\mathcal{D}$ **do**
**4**          run the Ising machine to obtain $\mathsf{E}_0(\theta^{(a)}, \Gamma^{(k)})$ and $\mathbf{z}^*$;
**5**      **end**
**6**      update $\Gamma^{(k)}, \lambda^{(k)}, \epsilon^{(k)}$ according to (12) - (13) - (14);
**7**   **end**
**8**   **return** $F_{\text{model}}(\theta) = F(\theta | \Gamma^{N_{\text{epochs}}}, \lambda^{N_{\text{epochs}}}, \epsilon^{N_{\text{epochs}}})$;

---

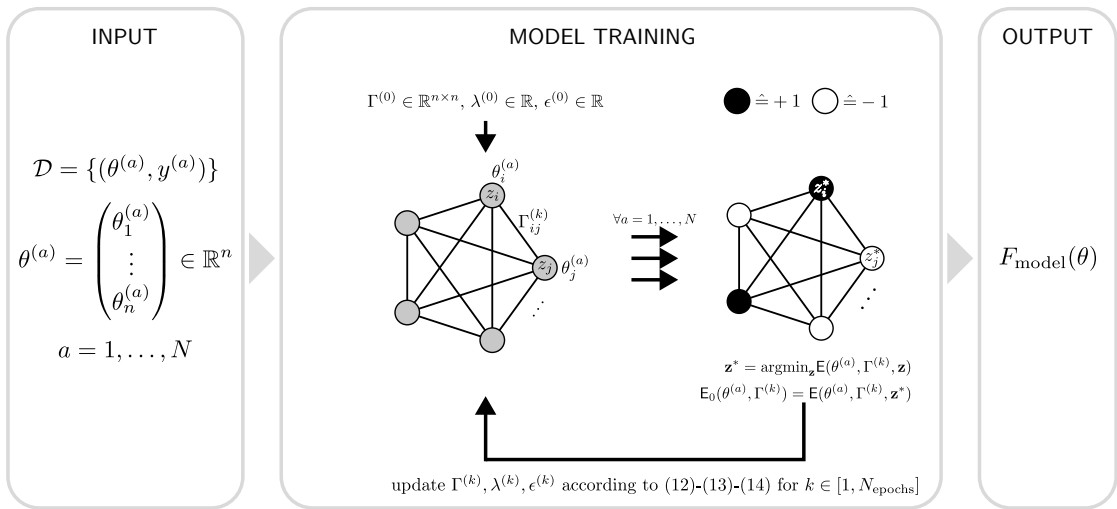

Figure 2: **Model training:** Illustration of the training process for the proposed model. In particular, given a dataset $\mathcal{D} = \{(\theta^{(a)}, y^{(a)})\}_{a=1,\dots,N}$, an Ising model is instantiated for each sample by setting the biases to $\theta^{(a)}$ and using the couplings $\Gamma$ as free parameters. Then, for each model, an Ising machine is run in order to obtain the spin configuration $\mathbf{z}^*$ and the corresponding model minimal energy $\mathsf{E}_0$. Finally, the collected values are used to update the couplings $\Gamma$ and the two additional parameters $\lambda$ and $\epsilon$ according to the rules presented in Theorem 3. This procedure is repeated $N_{\text{epochs}}$ times until the trained model $F_{\text{model}}(\theta) = F(\theta | \Gamma^{N_{\text{epochs}}}, \lambda^{N_{\text{epochs}}}, \epsilon^{N_{\text{epochs}}})$ is returned.

In this way, the model parameters can be optimized for a certain number of steps $N_{\text{epochs}}$. The complete training process is described as pseudocode in Algorithm 1 and illustrated as a flow diagram in Figure 2. In particular, for each training step $k$, the model is evaluated on each $(\theta^{(a)}, y^{(a)})$ pair in the training set $\mathcal{D}$ and the parameters are updated according to Theorem 3. The trained model is defined by the final iteration as

$$F_{\text{model}}(\theta) = F(\theta | \Gamma^{N_{\text{epochs}}}, \lambda^{N_{\text{epochs}}}, \epsilon^{N_{\text{epochs}}}). \tag{20}$$

Therefore, the training process bears similarities to that of a neural network but with a noteworthy distinction. Indeed, in our model, the conventional backpropagation step for calculating the partial derivatives is replaced by the Ising machine computation of $\mathsf{E}_0$ and $\mathbf{z}^*$. In particular, we propose the usage of quantum annealing as a well-suited Ising machine, which serves a dual purpose: executing the model according to (9) and facilitating the model training

through the iterative assessment of the loss function gradient. In detail, the spin configuration $\mathbf{z}^*$, retrieved from the annealer and representing the ground state of the qubit network, can be used to compute the parameter adjustments according to (12), (13) and (14). Instead, the corresponding energy value is used to compute the model prediction.

This ability to utilize the output of the Ising machine to train and evaluate the model constitutes the major distinction to other Ising machine-based models [33, 34] that require an explicit calculation of the corresponding derivatives to update the model parameters.

A model trained in this manner possesses the capability to predict inputs beyond those present in $\mathcal{D}$. Analogously to other machine learning models, this rests upon the expectation that, if the model is trained on an extensive dataset, it can assimilate and generalize from those examples, ultimately serving as an approximation of the original function within a certain value range. Moreover, although the Ising energy (4) depends only linearly on the input vector $\theta$, determining the minimum energy entails a complex interplay between the input and the model parameters $\Gamma$. Consequently, an open theoretical question regarding the class of functions that can be approximated through the proposed methodology arises. In other words, given an Ising model, what is its expressibility in terms of ground state energies by varying only the qubit couplings? From a practical perspective, the limitations of the quantum annealer architecture (number of qubits, topology connectivity, value bounds for $\theta$ and $\Gamma$) impose additional obvious constraints.

## 3.3 Hidden spins

In the proposed model, assuming a complete topology graph, the number of tunable parameters $\Gamma_{ij}$ scales quadratically with respect to the input dimension $n$. In practice, the number of model parameters is intrinsically fixed by the input dimensionality, akin to a neural network featuring only input and output layers. In the neural network scenario, to enhance the model expressiveness, the number of parameters is typically augmented by introducing additional hidden layers. In a similar way, we consider additional *hidden spins*, represented by additional nodes in the topology graph. These additional spins increase the number of couplings and, therefore, the number of parameters of the model. This is accomplished by adding a preprocessing step,

$$h_{\text{pre}} : \mathbb{R}^n \to \mathbb{R}^{n_{\text{total}}}, \tag{21}$$

mapping the original input vector $\theta$ from the feature space $\mathbb{R}^n$ to a higher-dimensional space characterized by $n_{\text{total}} = n + n_{\text{hidden}}$ dimensions, with $n_{\text{hidden}}$ representing the number of additional hidden spins. An illustration of this preprocessing step and the increase in the number of coupling parameters is given in Figure 3.

The preprocessing step does not affect the training process. Indeed, the model can still be trained as described in Section 3.2. Instead, the choice of the preprocessing function exerts a significant influence on the model's performance. For instance, let us consider a trivial preprocessing procedure that appends zero values to the input vector in order to reach the desired dimension. Although this approach would increase the number of model parameters, the hidden spins would be indistinguishable from each other, resulting in a very similar learning behavior and making them redundant. In contrast, initializing the additional dimensions with random values would mitigate this issue, but these values may overshadow the original input, especially if $n_{\text{hidden}} \gg n$. In this work, we propose and evaluate a first simple scheme to initialize additional spins based on a constant real-valued offset. This *offset initialization*

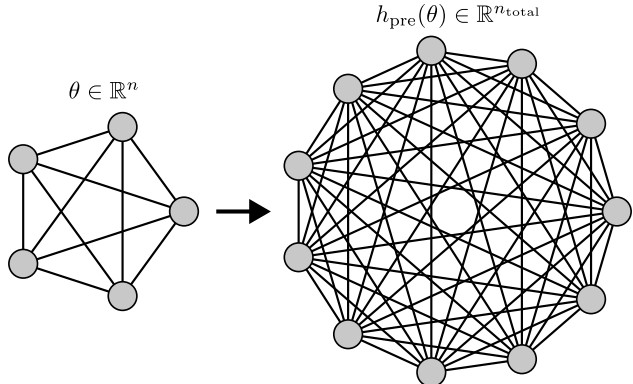

Figure 3: **Hidden spins:** Two exemplary Ising models with full connectivity. This comparison shows the increase in trainable coupling parameters (graph edges) when the original input $\theta$ is mapped to a higher dimensional space using a preprocessing step $h_{\text{pre}}$.

approach is defined as

$$\theta \in \mathbb{R}^n \rightarrow h_{\text{offset}}(\theta) = \begin{pmatrix} \theta \\ \theta + 1 \cdot d \\ \vdots \\ \theta + (l-1) \cdot d \end{pmatrix} \in \mathbb{R}^{n_{\text{total}}}, \tag{22}$$

where $d \in \mathbb{R}^n$, $l \in \mathbb{Z}^+$, and $n_{\text{total}} = ln$ (i.e., $n_{\text{total}}$ is a multiple of $n$). This corresponds to a repeated concatenation of the original input $\theta$ with an increasing real-valued offset $d$.

## 3.4 Computational cost

To define the computational cost of the proposed machine-learning model, three aspects must be taken into consideration: the initialization of the model, including the embedding into the available Ising machine; the training of the model, with repeated calls to the Ising machine and the update of the parameters; the evaluation on the provided input. In the following, the space and time complexity of the proposed model are discussed.

The encoding of the problem requires $n_{\text{total}}$ spin variables, considering the possibility of additional hidden spins (see Section 3.3). However, to achieve an all-to-all connectivity on sparse topology graphs like those of the D-Wave machines, an additional quadratic overhead has to be paid [35], resulting in a total space complexity of $\mathcal{O}(n_{\text{total}}^2)$.

The time complexity of the initialization (including the embedding) is $\mathcal{O}(n_{\text{total}}^2)$ [35]. Instead, the training of the model requires $N_{\text{epochs}}$ optimization steps. Specifically, in each optimization step, the Ising machine is evaluated on $N$ samples and the $\mathcal{O}(n_{\text{total}}^2)$ model parameters are updated. To provide its output, the Ising machine has to solve the spin glass system of the model (Equation 1), which is an NP-hard problem in general [36, 37]. Even for quantum annealers, the time required to find an exact solution is expected to be inversely proportional to the minimum energy gap of the ground state [23], resulting in an exponential worst-case complexity [36]. Nevertheless, an approximate solution can be found by leveraging the non-adiabatic system evolution [23]. Therefore, we can assume the Ising machine runtime to be upper bounded by some $\tau$ throughout the whole training process. Given the Ising machine outputs, the parameters update can be done in time $\mathcal{O}(n_{\text{total}}^2 N)$ using Theorem 3. Overall, the time complexity of the model turns out to be $\mathcal{O}(n_{\text{total}}^2 + N_{\text{epochs}} N(\tau + n_{\text{total}}^2) + \tau)$, where the last term corresponds to the evaluation of the model using a single Ising machine execution.

# 4 Empirical evaluation

This section provides an initial proof of concept of the model's capabilities. Indeed, this is neither a benchmarking exercise nor an in-depth analysis of the model's expressiveness but a demonstration of possible use cases and applications of the model. A detailed performance evaluation of the model, entailing the necessary statistical repetitions and the comparison to alternative models, is left for future work. To simplify the usage of the model, a Python package that automates the repeated calls to the Ising machines during the training of the model and also facilitates the cross-usage with other common Python machine learning packages (such as PyTorch) was published on Github [38]. As a first experiment, the model has been trained on randomly sampled datasets to demonstrate the trainability of the model itself according to the update rules of Theorem 3. Then, as real-world demonstrations, the model has been trained for the function approximation task and also as a binary classifier for the bars and stripes dataset.

## 4.1 Experimental setup

As discussed in Section 3, the model supports different Ising machines. In this work, we have considered simulated annealing and quantum annealing, both provided by the D-Wave Ocean Software SDK [39]. While the former represents a software implementation of simulated annealing, the latter directly accesses the superconducting annealing hardware supplied by D-Wave. In particular, the *Advantage_system5.4* has been used here. More in detail, the quantum annealing hardware in question is characterized by 5760 qubits and is based on the Pegasus topology, with an inter-qubit connectivity of 15. To control the hardware, D-Wave provides the Ocean SDK, which includes multiple software packages facilitating the handling of the annealing hardware. Among them, it is worth mentioning the *minorminer* package, which has been used to embed the problems into the annealer topology. In practice, to achieve the desired connectivity (all-to-all in this case), multiple physical qubits are chained together to form logical qubits; the drawback lies in the reduced number of available qubits. In particular, in each run, the embedding has been computed once for a fully connected graph of the required size and reused in the subsequent calls to the annealer; for this aim, the *FixedEmbeddingComposite* class of the Ocean SDK has been employed. Regarding the actual annealing process, the default setup has been used, namely, automatic rescaling of bias and coupling terms to fit the available hardware ranges, chain strength settings according to *uniform_torque_compensation*, an annealing time of $20\mu s$, and a twelve-point annealing schedule. To account for the high number of calls to the annealing hardware throughout training and save hardware access time, a number of reads (sampling shots) equal to 1 has been used for each annealing process. For more information, refer to Zenodo [40], where the set of notebooks used has been made available.

Concerning the model parameters, in all experiments, the couplings $\Gamma_{ij}^{(0)}$ have been initialized to zero and updated according to (12). Instead, $\lambda$ and $\epsilon$ have been kept fixed throughout the training process and considered as hyperparameters to facilitate the learning process. Specifically, the selection of the $\lambda$ value has been done manually to ensure that the model output was reasonably well-aligned with the range of values of the training data. By contrast, the $\epsilon$ value has been set according to the outcomes of the first round of sampling. In detail, the following rule has been used:

$$\epsilon = \frac{1}{N}\sum_{a=1}^{N}\left[y^{(a)} - F(\theta^{(a)}|\Gamma^{(0)}, \lambda, 0)\right] = \frac{1}{N}\sum_{a=1}^{N}\left[y^{(a)} + \lambda\sum_{i=1}^{n}|\theta_i^{(a)}|\right], \tag{23}$$

with the last equivalence being valid only if $\Gamma_{ij}^{(0)} = 0$ for $i, j \in \{1, \ldots, n\}$.

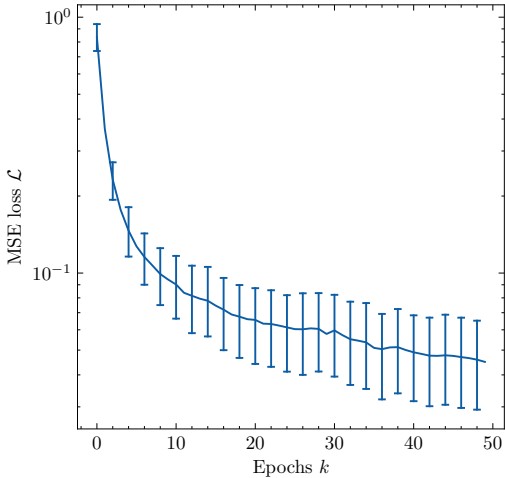

Figure 4: **MSE loss on random data:** Mean squared error, averaged over 30 randomly generated training sets of size $N = 20$. The MSE loss is tracked as a function of the number of epochs (with $N_{\text{epochs}} = 50$). The Ising machine in this experiment is the simulated annealing algorithm bundled in the Ocean SDK. The decreasing trend of the loss demonstrates the trainability of the model.

## 4.2 Random data

To demonstrate the trainability of the model, 30 distinct datasets, each comprising $N = 20$ data points with input dimension $n = 10$, have been considered. In particular, the input and target output values have been randomly sampled from a uniform distribution over the interval $[-1, 1]$. In addition, in this experiment, the simulated annealing algorithm bundled in the Ocean SDK has been employed as the Ising machine for estimating the ground state and the corresponding energy value. Hence, no quantum annealing hardware has been used in this case. The parameters used for simulated annealing can be found directly in the source code at [40]. Instead, regarding the parameters of the proposed model, $\lambda$ has been set to 1, and $\epsilon$ has been set according to (23) (taking a different value for each dataset). For the training process, $N_{\text{epochs}} = 50$ epochs have been executed, with $\eta = 0.2$. The MSE loss progression through the training is shown in Figure 4, where the error bars represent the standard deviation across the datasets.

Although this particular example lacks practical significance, it serves as a simple demonstration that the proposed Ising-machine-based parametric model can be effectively trained by utilizing its own output according to the update rules presented in Theorem 3. Furthermore, it highlights the fact that the discontinuity observed in the derivative of the optimal spin configuration $\mathbf{z}^*$, as discussed in the proof of Theorem 3, does not hinder the model's ability to minimize the loss function. In essence, the assumption made in (15) regarding the computation of the partial derivatives proves to be sufficiently accurate.

## 4.3 Function approximation

In this second experiment, datasets comprising $N = 20$ data points sampled from polynomial functions have been considered. Due to the limited quantum annealing time available on the D-Wave hardware, the analysis has been limited to two straightforward cases, and no statistical repetition has been performed. Although this shortage prohibits any general conclusion on the model's performance, it serves as a first demonstration of the possibility of using the model to approximate simple functions. Specifically, the following two polynomial functions of first

Table 1: Parameters used to train the model for the function approximation task.

|  | $n_{\text{total}}$ | $d$ | $\lambda$ | $\epsilon$ | $N_{\text{epochs}}$ | $\eta$ |
|---|---|---|---|---|---|---|
| $f_{\text{lin}}$ | 50 | 0.8/50 | −0.3 | −9.30 | 200 | 0.02 |
|  | 150 | 0.8/150 | −0.1 | 17.63 | 200 | 0.02 |
| $f_{\text{quad}}$ | 50 | 1/50 | −0.05 | −2.70 | 200 | 0.25 |
|  | 150 | 1/150 | −0.0167 | −4.23 | 200 | 0.25 |

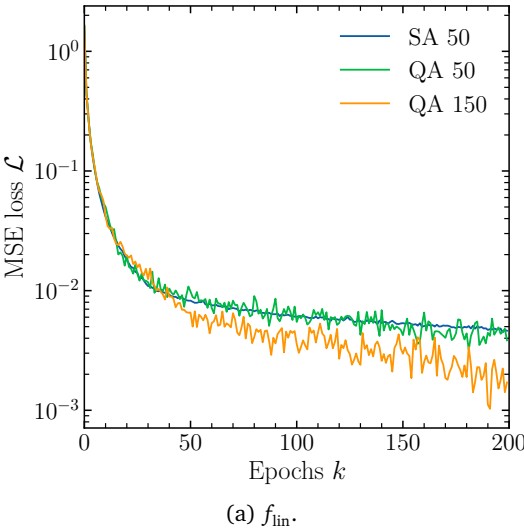

(a) $f_{\text{lin}}$.

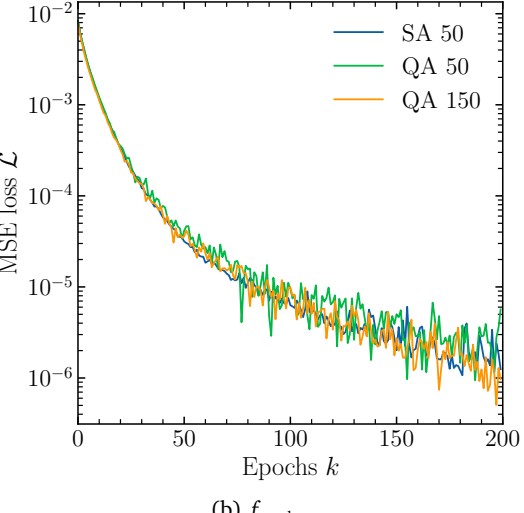

(b) $f_{\text{quad}}$.

Figure 5: **MSE loss in function approximation:** Evolution of mean squared error loss during training for linear **(a)** and quadratic **(b)** functions. The results achieved by both simulated annealing (SA) and quantum annealing (QA) are shown, with the numeric value following the method name representing the total number of hidden spins $n_{\text{total}}$. SA and QA perform similarly with equal sizes, with the fluctuations of QA being caused by the very low number of reads (1). For $f_{\text{lin}}$, a larger number of hidden spins corresponds to better performance of QA.

and second degree, respectively, have been considered:

$$f_{\text{lin}}(x) = 2x - 6,\tag{24}$$

$$f_{\text{quad}}(x) = 1.2(x - 0.5)^2 - 2.\tag{25}$$

In both cases, the coefficients have been chosen manually and arbitrarily, and the input domain has been restricted to the interval $[0,1]$. As the input dimensionality is $n = 1$, additional $n_{\text{hidden}}$ hidden spins (see Section 3.3) have been considered. In particular, two different total sizes $n_{\text{total}} = \{50, 150\}$ have been analyzed in order to study the effect of the number of hidden spins on the model learning. Additionally, the spins have been initialized using the offset technique described in Section 3.3. Regarding the model parameters, fixed values have been manually chosen for the scaling factor $\lambda$, whereas the offset $\epsilon$ has again been set according to (23). All model parameters used for the two total sizes considered are summarized in Table 1. In this case, simulated and quantum annealing have been employed as Ising machines and compared. The simulated annealing parameters are the same as those used in Section 4.2.

The MSE loss throughout the training epochs for the two functions is shown in Figure 5. In the case of the linear function (Figure 5a), the model demonstrates a significant reduction

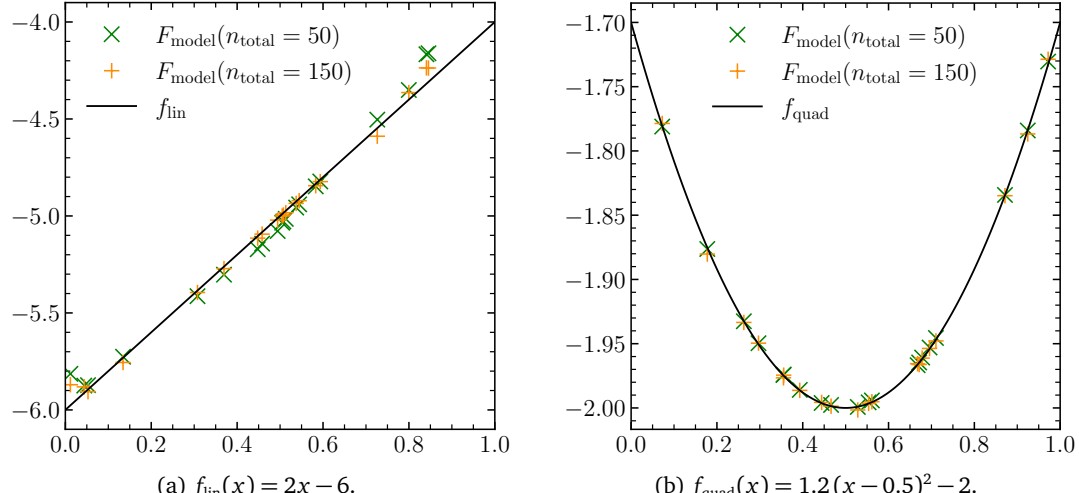

(a) $f_{\text{lin}}(x) = 2x - 6.$        (b) $f_{\text{quad}}(x) = 1.2(x - 0.5)^2 - 2.$

Figure 6: **Trained model output:** Output of the trained model $F_{\text{model}}$ compared to the original function (black line). In both cases (linear and quadratic), for both $n_{total}$ values, the model demonstrates the ability to approximate the function with good accuracy, performing slightly worse for $f_{\text{lin}}$, especially toward the edges of the considered interval.

in the mean squared error (MSE), over nearly three orders of magnitude, after approximately 200 optimization steps. Instead, in the case of the quadratic function (Figure 5b), the initial loss was already low, indicating that the offset method chosen for the hidden layers was appropriate for this dataset. Nevertheless, the model has managed to decrease the loss by nearly additional three orders of magnitude. It is also worth noting that, in both cases, for equal model sizes, the results achieved using the quantum annealing hardware align closely with those obtained by employing the simulated annealing algorithm. Specifically, the fluctuations in the quantum annealing loss are caused by the very low number of reads (1), resulting in non-optimal solutions occasionally returned by the annealer. Finally, the higher number of hidden spins (150) has shown significant advantages only for the linear function.

Instead, Figure 6 displays the output of the trained models compared to the original functions. It is clear that the model has successfully learned to approximate the target functions. Specifically, as expected from the low final loss value, the model closely aligns with the original function in the case of the quadratic function. Instead, in the linear case, the model performance deteriorates significantly toward the interval edges, and the output values exhibit a tendency toward a shape resembling an even-degree polynomial, especially for the case with less hidden spins ($n_{\text{total}} = 50$). This behavior stems from the initialization method chosen for the hidden spins and the symmetry properties of the Ising model. At extreme bias values, located near the interval boundaries, the biases exert a dominant influence on the energy term in Equation (4), causing $F(\theta) \rightarrow \infty$ as $|\theta| \rightarrow \pm\infty$. Consequently, the behavior resembles that of even polynomials, thus explaining the outliers in Figure 6a. Using more hidden spins ($n_{\text{total}} = 150$) reduces this effect by providing more trainable parameters to the model. It is also worth mentioning that different initialization methods for the hidden spins (e.g., taking the inverse values) influence this behavior.

## 4.4 Bars and stripes

In this last experiment, the proposed model has been applied to a different machine-learning task: binary classification. For this purpose, the well-known bars and stripes (BAS) dataset

has been used. In detail, the dataset consists of square matrices with binary entries such that the values in the rows/columns are identical within each row/column; the resulting patterns can be identified as bars/stripes, giving the dataset its name. Actually, the cases in which all entries of the matrix are the same have been left out as the label is not unique. Some examples are shown in Figure 7. Regarding the classification task, it consists in assigning a label $l \in \{\text{bars}, \text{stripes}\}$ to each matrix, corresponding to the pattern it represents. In particular, the dataset was created by randomly deciding the label of each data point and randomly assigning one of the two binary values to each row/column. This procedure has been repeated $N$ times, without accounting for duplicates.

In order to apply the proposed model to the BAS dataset, the input matrices have been flattened row-wise, and the binary values have been directly provided as input to the model. The binary labels $l \in \{\text{bars}, \text{stripes}\}$ have been encoded into $y$ and decoded from the model output $F_{\text{model}}$ according to

$$y = \begin{cases} 0, & l = \text{bars}, \\ 10, & l = \text{stripes}, \end{cases} \qquad l_{\text{model}} = \begin{cases} \text{bars}, & F_{\text{model}} \leq 5, \\ \text{stripes}, & F_{\text{model}} > 5, \end{cases} \tag{26}$$

with the factor 10 being arbitrarily chosen (different values can be used, but the $\lambda$ and $\epsilon$ parameters must be adjusted accordingly). For the training, a randomly generated dataset comprising $N = 80$ data points, with each data point representing a BAS matrix of size $12 \times 12$, has been used. In particular, the model has been trained for $N_{\text{epochs}} = 8$ epochs, with $\eta = 0.02$, and has been evaluated on a separate test set consisting of other 80 data points. Since no additional hidden spins have been employed, $n = n_{\text{total}} = 144$ in this case. Concerning $\lambda$ and $\epsilon$, the former has been manually set to $\lambda = -0.3$, while the latter has been set to $\epsilon = -15.43$ according to (23). Due to the large number of spins $n_{\text{total}} = 144$, only the quantum annealing hardware was used to train the model.

The results obtained are shown in Figure 8. Specifically, Figure 8a displays the model output during training for the training set and test set, respectively. The values shown are the average output values across all the data points with the same label, with the corresponding standard deviations indicated by the transparent envelopes. The dotted horizontal line represents the classification threshold from (26). In practice, the average output value for the two labels diverges, approaching 0 and 10, respectively, as the number of epochs increases. This means that the model has learned to increase the output value for stripe data points and lower it for samples labeled as bars. This generalizes also to the unseen examples of the test set, but the separation between the two classes is more marked for the training set. This effect is also visible in Figure 8b, where the MSE loss for the training set and test set is shown. In detail, the training loss decreases in a monotone way, while the test loss stagnates after a few epochs. This is a typical indicator of model overfitting, which could be addressed in different ways, among which increasing the number of training samples $N$ in order to help the model generalize. A similar conclusion can be drawn considering the accuracy of the model shown in Figure 8c. The trained model is able to correctly classify 79 out of 80 training samples, but the accuracy on the test set saturates at only about 75%.

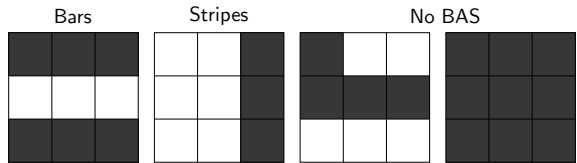

Figure 7: **Bars and stripes (BAS) dataset:** Illustration of exemplary $3 \times 3$ BAS and non-BAS data samples. The last two samples cannot be uniquely classified as bars or stripes and, therefore, are not part of the BAS dataset.

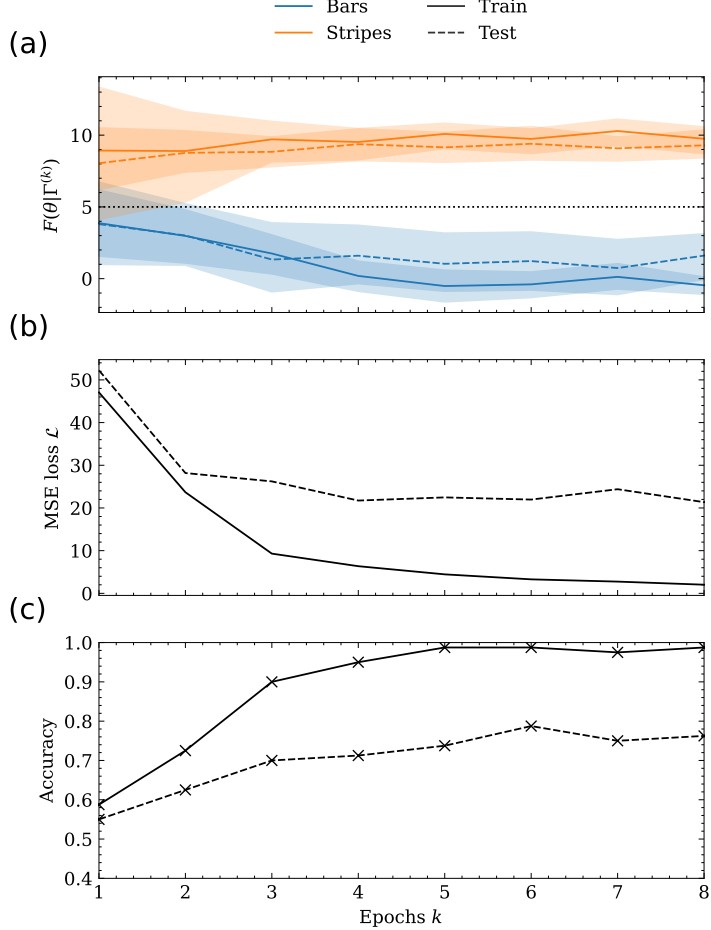

Figure 8: **Results on BAS dataset:** **(a)** Average model output value $F(\theta|\Gamma^{(k)}, \lambda, \epsilon)$ across all the data points with the same label $l \in \{\text{bars}, \text{stripes}\}$. The training (solid lines) and test (dashed lines) sets are considered independently; the envelopes represent the standard deviations, and the dotted horizontal line corresponds to the classification threshold according to (26). During training, the model learns to separate the two classes by increasing the energy for the stripes and decreasing it for the bars. **(b)** MSE loss for the training and test sets throughout epochs. The decreasing losses denote successful training, but the test loss stagnating after some epochs implies overfitting. **(c)** Accuracy for the training and test sets throughout the training. The accuracy on the training set reaches almost 1, with only one misclassified sample, while the accuracy on the test set also increases but saturates at about 75%.

In conclusion, this experiment has demonstrated the possibility of using the proposed model to address also binary classification tasks by choosing an appropriate encoding-decoding procedure for the model input and output. Indeed, the model has proven to be able to generalize to unseen examples while exhibiting overfitting effects, at least for the chosen dataset.

## 4.5 Choice of hyperparameters

Selecting appropriate values for the model's hyperparameters is a common issue in machine learning. Multiple hyperparameters have been manually set in the experiments presented in this work. These include the learning rate $\eta$, the number of epochs $N_{\text{epochs}}$, the problem encoding (see 26), the Ising machine parameters like the number of samples per step for simu-

lated annealing or the embedding procedure, the annealing time, and the number of reads for quantum annealing. Choosing appropriate values may reduce, for example, the fluctuations observed in Figure 6a. The values used here have been selected based on observations resulting from trial and error runs; the analysis of different configurations and a more systematic approach to choosing appropriate values are left for future work.

Among the model-related hyperparameters, the choice of the initialization strategy for the additional hidden spins has a significant impact. Specifically, when the input dimension is low, a large number of hidden spins $n_{\text{hidden}} \gg n$ may be necessary in order to have enough trainable model parameters. However, particular care must be put in choosing the corresponding new bias terms. Indeed, in preliminary experiments, it has been observed that initializing the biases in the wrong way may negatively affect the performance to the point that the model is unable to approximate the target function. Finding suitable ansätze for different tasks is still an open question.

## 5 Conclusion

In this paper, we have proposed a novel parametric learning model that leverages the inherent structure of the Ising model for training purposes. We have presented a straightforward optimization procedure based on gradient descent and we have provided the rules for computing all relevant derivatives of the mean squared error loss. Notably, if the Ising machine is realized by a quantum platform, our approach allows for the utilization of quantum resources for both the execution and the training of the model. Experimental results using a D-Wave quantum annealer have demonstrated the successful training of our model on simple proof-of-concept datasets, specifically for linear and quadratic function approximations and binary classification. This novel approach unveils the potential of employing Ising machines, particularly quantum annealers, for general learning tasks. In addition, it raises intriguing theoretical and practical questions from both computer science and physics perspectives. From a theoretical standpoint, questions regarding the expressibility of the Ising model arise, as well as inquiries into the classes of functions that the model can represent. These questions are non-trivial due to the non-linear minimization step involved. From a practical point of view, given the broad definition of the model and its similarity to other classical parametric models, a wide range of machine learning tools and methods can be explored to enhance its training. Advanced gradient-based optimizers and general learning techniques such as mini-batching, early stopping, and dropout, among others, offer promising avenues for improvement.

In addition to function approximation and binary classification, we aim to investigate the application of the model to other machine learning tasks, especially tasks in which the feature space is large, to reduce the necessity of additional hidden spins. This study might be extended with a comparison to other Ising machine-based models advancing the field of parametric machine learning models utilizing Ising machines.

## Acknowledgements

**Funding information** This work was partially supported by project SERICS (PE00000014) under the MUR National Recovery and Resilience Plan funded by the European Union - NextGenerationEU. E.Z. was supported by Q@TN, the joint lab between University of Trento, FBK-Fondazione Bruno Kessler, INFN-National Institute for Nuclear Physics and CNR-National Research Council. The authors gratefully acknowledge CINECA for providing computing time on the D-Wave quantum annealer within the project "Testing the learning performances of

quantum machines", and the Jülich Supercomputing Center for providing computing time on the D-Wave quantum annealer through the Jülich UNified Infrastructure of Quantum computing (JUNIQ).

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
