# Peer review of "A general learning scheme for classical and quantum Ising machines"

_SciPost Physics Core, doi:SciPost Phys. Core 7, 013 (2024)_

## Round 1 · Referee Report · Anonymous (Referee 1) · 2024-1-31

Strengths

1- The paper introduces an innovative machine learning model centered around Ising machines, showcasing a new perspective in the field.

2-The report is well-written and structured, effectively conveying complex concepts in a clear and comprehensible manner.

3-The proposed model demonstrates versatility by addressing tasks such as function approximation and binary classification.

Weaknesses

1- The report acknowledges the lack of a comprehensive performance evaluation, including statistical repetitions and comparisons to alternative models. A more thorough evaluation would strengthen the robustness of the findings.

2- While the paper provides a proof of concept, the deferral of statistical analysis to future work leaves some uncertainty about the model's performance and generalizability.

3- The paper touches on the model's capabilities in tasks such as function approximation and binary classification but does not extensively explore its variability or limitations in handling more complex scenarios.

4- The report could benefit from a discussion on the computational cost associated with the proposed model, especially when using quantum annealers, providing a more comprehensive perspective for readers.

Report

The paper, titled "A General Learning Scheme for Classical and Quantum Ising Machines," delves into an innovative machine learning model centered around Ising machines. Drawing inspiration from the training of Boltzmann machines, the authors establish a supervised learning model for Ising machines. This model demonstrates trainability through gradient descent on a mean squared error loss function.

The approach introduces a universal neural framework, where data is represented by spins' biases, and parameters act as weights between spins, applicable to both classical and quantum machines. In this model, the traditional backpropagation step for partial derivatives calculation is substituted by the Ising machine's computation of ground state E0 and the corresponding spin configuration z ∗.

Experimental results using a D-Wave quantum annealer showcase the model's prowess, addressing tasks such as function approximation and binary classification. Simulated annealing and quantum annealing serve as Ising machines, illustrating the model's successful training on uncomplicated datasets.

The paper concludes by exploring theoretical questions about the Ising model's expressibility and proposing practical avenues for enhancing the model's training using diverse machine learning tools.

While the paper provides a proof of concept for the model's capabilities, a comprehensive performance evaluation involving statistical repetitions and comparisons to alternative models is deferred for future work.

In summary, the paper introduces a new method for parametric learning through Ising machines, particularly quantum annealers, in the realm of general learning tasks.

This approach invites further exploration, as the authors emphasize potential applications and comparisons with other Ising machine-based models.

Requested changes

(Optional) The report could benefit from a discussion on the computational cost associated with the proposed model, especially when using quantum annealers, providing a more comprehensive perspective for readers.

---

## Round 2 · Referee Report · Anonymous · 2024-2-26

Report

The current form of the work meets the acceptance requirements of SciPost, therefore I recommend its publication

---

## Round 2 · Author Response

Dear Editors and Reviewers,

Many thanks for the thorough consideration of our manuscript, the positive feedback, and the provided suggestions.

In the following comments on the feedback of Report 1 indicated by ">>>><<<<" and “**** Response:” respectively. When referring to changes in the manuscript, the corresponding line numbers are mentioned, and the changes are highlighted in the provided pdf in blue and sans-serif.

Strengths 1- The paper introduces an innovative machine learning model centered around Ising machines, showcasing a new perspective in the field. 2-The report is well-written and structured, effectively conveying complex concepts in a clear and comprehensible manner. 3-The proposed model demonstrates versatility by addressing tasks such as function approximation and binary classification. <<<<

**** Response: Thank you for the nice summary of our work and the positive review.

Weaknesses 1- The report acknowledges the lack of a comprehensive performance evaluation, including statistical repetitions and comparisons to alternative models. A more thorough evaluation would strengthen the robustness of the findings.

2- While the paper provides a proof of concept, the deferral of statistical analysis to future work leaves some uncertainty about the model's performance and generalizability. <<<<

**** Response 1 + 2: As correctly indicated, the current evaluation lacks the necessary number and broadness of example runs to offer the statistical weight necessary to comment on the model performance. This is due mainly because of the limited access to computational time on the quantum hardware, and we plan to address this point in future work.

3- The paper touches on the model's capabilities in tasks such as function approximation and binary classification but does not extensively explore its variability or limitations in handling more complex scenarios. <<<<

**** Response 3: Due to the very general definition of the proposed model, it can be applied to almost arbitrary machine-learning tasks. Similar to the early days of neural networks, it is currently unclear in which cases the model performs well/poorly. Exploring these possible applications basically opens a new research direction, which will be addressed in future work.

4- The report could benefit from a discussion on the computational cost associated with the proposed model, especially when using quantum annealers, providing a more comprehensive perspective for readers. <<<<

**** Response 4: We thank the referee very much for pointing this out. We fully agree and added an additional section, 3.4, “Computational cost” (l.324-347), addressing this issue. Both for general Ising machines, but also in regard of quantum annealers and their classical counterparts.

Overall, thank you for your careful consideration and best regards,

The authors.

---

## Round 2 · List of Changes

- added section 3.4, “Computational cost” (l.324-347): We added a subsection discussing the computational cost to initialize, train, and evaluate the proposed model. Predicting the runtime of the Ising machine (in particular, the quantum annealer) is a highly non-trivial task as the Ising model to be solved changes throughout training, depending on the parameter updates. Nevertheless, we discussed different cases and added references that discuss in detail the runtime of quantum annealers in comparison to their classical alternatives, providing the reader with a more comprehensive overview.

- minor grammatical and spelling errors.

---

## Editorial Decision

published